# Hyaluronic Acid: A Powerful Biomolecule with Wide-Ranging Applications—A Comprehensive Review

**DOI:** 10.3390/ijms241210296

**Published:** 2023-06-18

**Authors:** Giorgia Natalia Iaconisi, Paola Lunetti, Nunzia Gallo, Anna Rita Cappello, Giuseppe Fiermonte, Vincenza Dolce, Loredana Capobianco

**Affiliations:** 1Department of Biological and Environmental Sciences and Technologies, University of Salento, 73100 Lecce, Italy; giorgianatalia.iaconisi@unisalento.it (G.N.I.); paola.lunetti@unisalento.it (P.L.); 2Department of Engineering for Innovation, University of Salento, 73100 Lecce, Italy; nunzia.gallo@unisalento.it; 3Department of Pharmacy, Health, and Nutritional Sciences, University of Calabria, 87036 Arcavacata di Rende, Italy; annarita.cappello@unical.it; 4Department of Biosciences, Biotechnologies and Biopharmaceutics, University of Bari, 70125 Bari, Italy; giuseppe.fiermonte@uniba.it

**Keywords:** hyaluronic acid, biomaterial, bioactive molecule, metabolic pathways, bioproduction, fermentation processes

## Abstract

Hyaluronic acid (HA) is a glycosaminoglycan widely distributed in the human body, especially in body fluids and the extracellular matrix of tissues. It plays a crucial role not only in maintaining tissue hydration but also in cellular processes such as proliferation, differentiation, and the inflammatory response. HA has demonstrated its efficacy as a powerful bioactive molecule not only for skin antiaging but also in atherosclerosis, cancer, and other pathological conditions. Due to its biocompatibility, biodegradability, non-toxicity, and non-immunogenicity, several HA-based biomedical products have been developed. There is an increasing focus on optimizing HA production processes to achieve high-quality, efficient, and cost-effective products. This review discusses HA’s structure, properties, and production through microbial fermentation. Furthermore, it highlights the bioactive applications of HA in emerging sectors of biomedicine.

## 1. Introduction

HA is a linear glycosaminoglycan polymer present in the extracellular matrix of vertebrate tissues, including connective, epithelial, and neural tissues [1]. It is involved in several biological processes such as embryonic development, wound healing, cancer progression, angiogenesis, inflammation, and bone regeneration [2,3,4]. HA has been found to regulate essential cellular processes such as cell adhesion, proliferation, and differentiation [5,6].

In addition to belonging to vertebrate connective tissues, HA could be found in several bacterial strains (e.g., groups A and C streptococci and *Pasteurella multocida*) as a component of their capsules and mucus with the role of conferring adherence, protection, and molecular mimicry necessary to evade the host’s immune system during their infection process [7].

Over the past 20 years, HA has garnered an exponential interest due to the continuous scientific discoveries regarding its intrinsic properties and subsequent applications. Ongoing research continues to investigate the mechanisms underlying its biological activity.

HA applications cover several fields ranging from cosmetic formulations to viscosurgery, ophthalmology, orthopedic surgery, rheumatology, tissue regeneration, and targeted cancer therapy [8]. This broad range of applications is possible due to the specific bioactivity exhibited by HA based on its molecular weight (MW). Furthermore, in addition to the use of pure HA, an emerging trend is the combination of HA with other bioactive ingredients or materials for several applications such as facial volume restoration [9,10,11], osteoarthritis treatment [12,13,14] (NCT00653432, NCT05683327), cosmetic medicine [15,16,17,18,19], nasolabial fold reduction [20,21], cancer treatment [22,23,24,25] (NCT05024773), atherosclerosis diagnosis and treatment [26], psoriasis treatment [27,28], urinary tract infection treatment [29,30], tissue engineering [31,32,33,34], antioxidant and anti-inflammatory effects [35,36,37], and wound healing [38,39].

The aging population, the consequent increase in demand for antiaging products, and the rise in age-related diseases along with the introduction of technologically advanced products, the growing preference for non-invasive techniques, the adoption of aesthetic procedures, and the need for faster and natural-like results are all factors that have affected and will continue to impact the growth in research and the market size of HA in the coming years.

As a result, the global market for hyaluronic acid (HA) is expected to witness significant growth in the coming years, with a projected value of USD 16.8 billion by 2030 [40,41]. This increasing market share emphasizes the importance of optimizing HA production processes to meet high-quality standards, ensure efficiency, and maintain affordable costs.

After providing a brief overview of HA’s structure and properties, the state of the art in HA production methods is discussed. Particular attention is given to the current status and the role of HA as a bioactive molecule in the emerging fields of biomedicine. Preclinical and clinical progresses of the last 10 years are thoroughly discussed.

## 2. Methodology

A deep search was undertaken on studies about the current status and the role of HA as a bioactive molecule. The electronic search engines used were PubMed (https://pubmed.ncbi.nlm.nih.gov, accessed on 23 March 2023), ScienceDirect (https://www.sciencedirect.com, accessed on 4 January 2023), Google Scholar (https://scholar.google.com, accessed on 23 March 2023) and U.S. National Library of Medicine (https://clinicaltrials.gov, accessed on 30 April 2023). The keywords used were ‘hyaluronic acid’, ‘metabolic pathway’, ‘fermentation’, and ‘bioactive molecule’. Several synonyms were searched for each keyword (i.e., hyaluronan, hyaluronate, bioproduction, biosynthesis, and bioactive compound). The search included the recent updates (2013–2023) related to HA-based formulations (including clinical trials as well as in vitro and in vivo studies) independently from their level of evidence.

## 3. HA: Structure and Properties

HA, also known as hyaluronan, is an important member of the glycosaminoglycan (GAG) family characterized by high polymerization and a macromolecular structure. In marked contrast with the other GAG members (i.e., chondroitin sulfate, dermatan sulfate, heparan sulfate, and keratan sulfate), HA does not undergo sulfation or other modifications along its entire length. Moreover, it is not synthesized in the rough endoplasmic reticulum and Golgi apparatus but is produced on the cytoplasmic surface of the plasma membrane by hyaluronan synthase enzymes. Additionally, it has been documented that HA does not attach to a core protein as observed in other GAGs [42,43].

HA consists of repeating units of β-(1,4)-glucuronic acid (GA) and β-(1,3)-N-acetylglucosamine (NAG) that are repeatedly linked together by alternating β-1,3 and β-1,4 glycosidic bonds (Figure 1) [43].

Based on the number of repeating units, hyaluronic acid can be subdivided into three forms: low-molecular-weight (LMW) HA (6–200 kDa), medium-molecular-weight (MMW) HA (0.2–1.0 MDa) and high-molecular-weight (HMW) HA (>1 MDa). Generally, HA can consist of up to 10,000 or more repeating units, resulting in a molecular mass of approximately 4 million Da [44,45]. The structure of this molecule is energetically stable only if both sugars are in beta conformation. This conformation allows the anomeric carbon, hydroxyl group, and carboxylic group to occupy sterically favorable equatorial positions, while the hydrogens assume less favorable axial positions [44]. The presence of carboxylic groups in HA results in a negatively charged molecule that is highly hydrophilic at physiological pH. As a result, under these conditions, HA has the ability to bind a large amount of water molecules and become viscous through the establishment of intramolecular and intermolecular hydrogen bonds [41].

The physico-chemical properties of HA play a crucial role in maintaining the hydration of the extracellular matrix, where it acts as a water-binding component [46]. By performing this function, HA contributes to the resistance of tissues against physical stresses and it ensures the right moisture level [47].

Furthermore, HA exhibits a peculiar mesh structure that acts as a barrier against several exogenous substances, including bacteria and infectious agents (except *S. aureus* and *S. pyogenes*), by slowing down their filtration ability [48]. Additionally, HA regulates tissue homeostasis, and it is also involved in the growth of epithelial cells, eosinophils, and macrophages [49,50,51]. It also forms a peri-cellular lining around several cell types and thus, depending on molecular weight, it acts as a signaling molecule and a regulator of cell adhesion, migration, and proliferation [52].

As previously mentioned, the different functions of HA are closely linked to its molecular weight. Indeed, there is a correlation between HA bioactivity and its MW. Accordingly, the size of the molecule can affect its affinity for receptors and the possibility of forming receptor complexes. Additionally, the molecular size of HA plays a role in its cellular absorption [53].

Basically, the mechanical strength of hyaluronic acid (HA) increases with an increase in its molecular weight (MW). Consequently, HMW HA is preferably used for applications that require strong viscoelastic properties and mechanical strength, such as joint supplements and wound dressings [54]. HMW HA has also been found to possess anti-inflammatory and immunosuppressive functions and to have a role in angiogenic mechanisms [53,55]. Therefore, it is largely used as a space filler and a natural immunologic depressant [1,56,57].

Furthermore, HMW HA is a key component of synovial fluid, where it plays an essential role as joint lubricant [49], and therefore it is significant in many physiological and pathological conditions (as in the case of osteoarthritis). On the other hand, MMW and LMW HA have been found to stimulate the synthesis of heat-shock proteins thereby exhibiting proangiogenic, antiapoptotic, and immunostimulatory properties [55]. LMW HA in particular finds applications in topical cosmetical applications and plastic surgery thanks to its ability to permeate the epidermis stratum and improve tissue hydration and healing [56,58]. In particular, a study conducted by Essendoubi et al. (2016) demonstrated that LMW HA (20–300 kDa) is able to penetrate the stratum corneum of skin tissue [59].

It is worth noting that in various commercial products and production sources, in addition to HA, sodium hyaluronate is often found. It is the sodium salt form of HA and is commonly used in various applications due to its enhanced stability and solubility. Moreover, it has a smaller molecular size, which enables it to effectively penetrate the skin [60]. Furthermore, the salt form can be produced with a high purity rate [60,61].

## 4. HA Microbial Metabolic Pathways

Several studies on HA biosynthesis focused on natural producer microorganisms such as group A and C streptococci and *Pasteurella multocida*, which produce HA as a component of their capsules and mucus [62,63].

Generally, HA is synthesized via two different metabolic pathways, resulting in the production of two precursors: UDP-GA and UDP-NAG [64] (Figure 2).

Glucose is imported into the bacterial cell and phosphorylated by hexokinase, converting it into glucose-6-phosphate (glucose-6P) in the cytoplasm. Then, it is converted by the phosphoglucomutase in glucose-1-phosphate (glucose-1P), after which the UDP-glucose pyrophosphorylase catalyzes the formation of UDP-glucose. This UDP-glucose is subsequently oxidized by the UDP-glucose 6-dehydrogenase to obtain the first HA precursor: UDP-GA. However, other metabolic pathways compete with this conversion since the glucose-1P can be used for glycogen production [62] and the UDP-glucose can be converted to trehalose-6P [62]. These two metabolic pathways represent a metabolic limit for HA production [54].

The second HA precursor is produced through a second pathway in which glucose-6P is converted to fructose-6-phosphate (fructose-6P) by the phosphoglucoisomerase. UDP-glucose 6-dehydrogenase and phosphoglucoisomerase are the most important competing enzymes because the first one directs glucose-6P into the Pentose Phosphate Pathway, while the latter converts glucose-6P into fructose-6P via the Embden–Meyerhof–Parnas pathway [54]. Cells typically utilize the majority of glucose-6P for these two competing metabolic pathways rather than UDP-GA synthesis [65]. Furthermore, cells can use the phosphoglucoisomerase to convert fructose-6P into glucose-6P, which can be utilized in the Pentose Phosphate Pathway [62]. Then, glucosamine-6-phosphate (glucosamine-6P) is formed from fructose-6P by the action of amidotransferase. Glucosamine-6P is the substrate of the mutase that leads to the formation of the glucosamine-1-phosphate (glucosamine-1P).

Amidotransferase, on the other hand, competes with the UDP-NAG cascade by converting glucosamine-6P to fructose-6P [54]. Indeed, fructose-6P can be converted into fructose-1,6-bisphosphate through the ATP-dependent phosphofructokinase, which is utilized in the Embden–Meyerhof–Parnas pathway for energy generation and cell growth [54]. In addition, it must be considered that if cells have a limited amount of carbon sources to feed these two main metabolic pathways, a significant portion of cellular glucosamine-6-phosphate could be converted to fructose-6P and glucose-6P to meet the metabolic needs of cells [54]. Consequently, cells will have a limited reserve of glucosamine-6P to synthesize glucoamine-1P.

Glucosamine-1P is further acetylated and phosphorylated by pyrophosphorylase to form the second precursor: UDP-NAG. In this case, reduced flux to UDP-NAG due to competing pathways also is a significant limitation to HA synthesis [62]. Consequently, at the end of these metabolic pathways, UDP-GA and UDP-NAG are polymerized by the hyaluronic acid synthase (HAS), which catalyzes the formation of the HA chain [66]. The biosynthetic pathway of HA in bacteria shows that the HAS is a key enzyme that catalyzes the polymerization of HA chains. In particular, the streptococcal HAS is classified as a class I HAS because it is an integral membrane protein that catalyzes the addition of UDP-GA acid and UDP-NAG to the growing HA chain [67] (Figure 3A).

Conversely, the HAS of *P. multocida* belongs to class II because it is a membrane-associated glycosyl transferase. This enzyme also has two independent active sites: (i) an “A1” domain that performs the UDP-NAG transferase activity, and (ii) and an “A2” domain that carries out the UDP-GA transferase activity (Figure 3B). Therefore, HAS catalyzes the sequential elongation of a growing HA chain at the non-reducing end with its β-1,3-glucuronyl-transferase and β-1,4-*N*-acetylglucosaminil-transferase domains, which work synergistically as a single polypeptide [67].

Hence, it is clear that HA production is an energy- and carbon-intensive process for the bacterial cell. Its metabolic pathway consumes large quantities of its donor substates, thereby linking HA synthesis to glucose metabolism. As already mentioned, glucose-6P is an important substrate of the Pentose Phosphate Pathway, while fructose-6P is an important intermediate of the glycolytic pathway. Moreover, some intermediates of HA biosynthesis are also required for cell wall biosynthesis [68]. In particular, glucose-1P is an important wall polysaccharide, UDP-glucose is used by streptococci for teichoic acid biosynthesis [69], and finally NAG is a component of the structural unit of peptidoglycan [70].

It is easy to see that HA synthesis is strongly influenced by the cellular availability of the two precursors [71] and thus to cells’ metabolic needs. Moreover, it has been observed that if each precursor is not supplied at sufficient rates, the amount of the synthesized HA and its MW are reduced [72]. The carbon source must be shared equally between the two pathways to correctly produce HA. Due to this, several studies were conducted in order to maximize hyaluronic acid production.

## 5. HA Production Methods

HA can be obtained from different sources, including animal tissues and microorganisms. After Meyer and Palmer first isolated HA from bovine vitreous humor, it was also extracted from other tissues such as the umbilical cord, rooster combs, and synovial fluid [73]. However, HA from animal sources has sparked concerns due the high costs of the extraction process, low yields, and side effects. One of the main concerns is the possibility of immune responses and allergies triggered by the presence of other substances such as host proteins and DNA in animal-derived HA [42].

Additionally, it is important to note that animal tissues naturally contain hyaluronidases, enzymes that can degrade HA. As a result, a portion of the HA present in animal tissues may undergo degradation before the extraction process takes place. This degradation can reduce the overall yield and quality of the extracted HA. Therefore, the presence of hyaluronidases in animal tissues further contributes to the challenges and limitations associated with obtaining HA from animal sources [56]. Moreover, it should be taken into account that the partial degradation of HA is further promoted by the harsh conditions of the multistep extraction process [56].

In light of these challenges, alternative methods for industrial production of HA have been developed. In particular, some pathogen bacteria such as Pasteurella multocida and Gram-positive Streptococcus Group A and C were investigated as production systems because of their natural ability to produce and secrete HA as a secondary metabolite. Although they emerged as interesting alternatives [42], they presented several limitations such as the presence of bacterial endotoxins, which limited their biomedical applications [74].

After having identified genes involved in the biosynthetic pathway of hyaluronic acid, several bacterial species were employed to produce HA, including genetically engineered strains and generally recognized as safe (GRAS) microorganisms [56]. Thus, metabolic engineering provided valuable strategies to obtain HA from GRAS microorganisms because organisms that naturally produce HA are mostly pathogenic. Therefore, using techniques employed over the years for the expression of active proteins [75,76,77,78,79], HA has also been successfully produced in a wide range of heterologous hosts, including Lactococcus lactis [72,80], Escherichia coli [81], Bacillus subtilis [82], and Bacillus megaterium [81]. In particular, a recent study showed that *B. megaterium* is one of the most suitable host microorganisms for HA biosynthesis thanks to its great ability to secrete recombinant proteins in the culture medium [81]. Nevertheless, even in the case of genetically engineered GRAS microorganisms, the presence of bacterial proteins and nucleic acids could still constitute a limitation [81].

The advantages and limitations of HA extracted from different sources are summarized in Table 1.

## 6. HA Fermentation Process in Engineered Host Microorganisms

As previously mentioned, industrial and scientific research is looking for optimized HA production methods with controlled parameters, high yields, low production costs, a high MW, and a high purity of the final product. HA bacterial production through metabolic engineering represents a promising strategy. Over the last years, several microorganisms have been engineered in order to produce HA. Different fermentation methods and culture media have been tested. Not only natural microbial producers of HA have been engineered in order to optimize the characteristics of the final product, but non-natural producers such as GRAS microorganisms (i.e., *E. coli*, *B. subtilis*, and *B. megaterium*) have been as well.

First of all, several researchers attempted to maximize HA production by tuning the microorganism culture conditions rather than engineering them. Parameters such as the initial pH of the broth, temperature, time of incubation, and agitation speed were evaluated as important parameters that influenced the yield of HA. Among several bacteria, *S. equi* spp. *zooepidemicus* was the most investigated. Güngör et al. found that the growing conditions that allowed them to improve the HA final yield (12 g/L) were pH 7.8, an incubation temperature of 33 °C, an incubation time of 24 h, and an agitation speed of about 187 rpm [87]. In another recent study, UV-induced mutagenesis on *S. zooepidemicus* allowed the researchers to further increase the HA yield, which reached about 4.2 g/L after 36 h of fermentation [88]. In this case, the optimal fermentation was obtained by incubating it at 37 °C and pH 7.4 in a fermentation broth that contained glucose as the carbon source, phosphate as the essential nutrient, peptone 250 as the nitrogen source, and dibasic potassium phosphate as the phosphate source [88]. Then, to enhance the efficiency of the hyaluronic acid production, a semi-continuous fermentation process that consisted of two-stage 3–L bioreactors was developed, and recombinant hyaluronidase *SzHYal* (300 U/L) was added into the second stage bioreactor to reduce the broth viscosity. In this way, an HA yield of 30 g/L was successfully attained [88]. Another strategy showed that as an alternative to glucose, molasses could be used as a carbon source in the fermentation medium of *S. zooepidemicus* [89]. Indeed, the presence of this alternative and cheaper sugar allowed an increase in the yield of the HA up to 0.2 g/L compared to standard conditions (43 mg/L) [89]. In addition to the pH, the incubation temperature, agitation speed, soil composition, and size of inoculum are key parameters of the fermentation process that affect the yield of the final product. Some researchers found that a streptococcal inoculum size of 10% had the greatest effect on the fermentation process and also that an agitation speed of 300 rpm could increase the HA yield [90]. This may have occurred because at this rate of speed, a greater subdivision of air bubbles could occur with a consequent larger surface area for the gas–liquid mass transfer, which therefore led to a reduction in the thickness of the gas and liquid films responsible for the resistance to the mass transport [90]. In this way, it is easy to understand that several parameters must be optimized in order to enhance the microbial production of HA starting from natural producer microorganisms.

With the aim of reducing extracted product endotoxins, recombinant non-natural microbial producers were designed, developed, and investigated in depth [62]. A variety of organisms including *B. subtilis*, *B. megaterium*, and *E. coli* have all been genetically engineered in order to produce HA [81,91,92]. Genetically modified *E. coli* strains have several potential advantages when compared to *Streptococcus* strains and therefore offer a promising alternative [62]. Additionally, compared to *Streptococcus* strains, several *E. coli* strains are non-pathogenic [54], and they can be grown at a high cell density using cheap and simple media. As a result, the fermentative process of *E. coli* could be improved through metabolic engineering [93]. For these reasons, many studies have been carried out with recombinant *E. coli*. In particular, a recent study compared the HA production in different *E. coli* strains (pLysY/lq, Rosetta2, Rosetta2 (DE3) pLysS, and Rosetta-gami B (DE3) pLysS) and *B. megaterium* (MS941) cells [91]. Firstly, the researchers used different fermentation conditions for *E. coli* cells. Initially, they engineered the bacterial cells by transforming them with different plasmids. The first introduction of the *hasA* gene of *S. equi* spp. *zooepidemicus* in *E. coli* Rosetta2 cells allowed them to obtain a yield of 8.8 mg/L [91]. Then, the introduction of a second set of genes, which were *hasA*, *hasB* and *hasC* genes in *E. coli* pLysY/lq cells, allowed them to reach a yield of 208.3 mg/L (by adding MgCl_2_, K_2_HPO_4_, and sorbitol to the fermentation broth) [91]. Nasser et al. also attempted to obtain HA from Rosetta2 (DE3) pLysS, with a successful yields of 346.7 mg/L in Super Optimal broth with Catabolite repression (SOC) medium and 500 mg/L in terrific broth (TB) medium [91]. Lastly, they analyzed the ability of Rosetta-gami B (DE3) pLysS to produce HA by transforming it with *hasA* (hyaluronic acid synthase gene), *hasB* (UDP-glucose dehydrogenase gene), *hasC* (UDP-glucose pyrophosphorylase gene), *hasD* (acetyltransferase gene), *hasE* (phosphoglucoisomerase gene) genes; in this way, they obtained 585 mg/L of HA [91].

In this case, it is also possible to note that the composition of the fermentation medium and the bacterial strain affect the HA yield. In a study by Nasser et al., it emerged that *B. megaterium* was able to produce a higher amount of HA rather than *E. coli*. Indeed, by expressing only the *hasA* gene along with the addition of xylose, MgCl_2_, K_2_HPO_4_ and sorbitol to Luria-Bertani (LB) fermentation broth, they were able to obtain 50 mg/L of HA. By expressing *hasA*, *hasB*, and *hasC* with the addition of sucrose to LB fermentation broth, the yield of HA was about 2116.7 mg/L. In TB broth instead, the yield was decreased (2016 mg/L). At this point, they modified the fermentation medium, and they observed that in A5 + sodium 2-hydroxy-3-(morpholin-4-yl)propane-1-sulfonate (MOPSO) medium (containing yeast extract and a mineral medium based on MOPSO buffer), the amount of HA was about 1990 mg/L. As a final step, they introduced *hasA*, *hasB*, *hasC*, *hasD*, and *hasE* genes in *B. megaterium* cells. In LB medium with sucrose, the HA yield was about 2350 mg/L; in A5+MOPSO medium, the HA amount was about 2436 mg/L [91].

Another study used *E. coli* Rosetta (DE3) as the host to obtain the production of HA by optimizing the fermentation medium and inserting the *hasA* and *hasE* genes [91]. In this case, the fermentation broth contained glucose as the carbon source, MgSO_4_ as the Mg^2+^ donor, and KH_2_PO_4_ and K_2_HPO_4_ as the phosphate sources [91]. This study further demonstrated the effect of fermentation temperature on HA yield, since the maximum HA biosynthesis was observed when fermentation was run at less than 37 °C. Finally, it was observed that the gene expression inductor concentration also could affect the HA yield. The highest HA concentration was obtained when the cell growth was at the lowest; that is, when the concentration of the gene expression inductor was the highest. Indeed, in these conditions, the cellular growth decreased and the HA concentration increased [91].

As previously mentioned, bacilli also can be used for HA production since they are safe, can be easily genetically tractable, and are able to grow in simple culture media [94]. In particular, in a recent study *B. subtilis* 3NA was engineered by overexpressing the *hasA* gene of *S. zooepidemicus* and its other endogenous genes involved in the HA biosynthetic pathway [92]. The fermentation medium contained glycerol as the carbon and energy source, H_3_PO_4_ as the phosphate source, NH_4_OH as the nitrogen source, MgSO_4_ as the Mg^2+^ donor and sulfur source, and finally KOH to buffer the pH [92]. Hence, the obtained HA yield was comparable with the streptococcal one [92].

Thus, all mentioned works allowed us to understand that the strain, exogenous genes, and fermentation conditions (i.e., soil composition, temperature, pH, concentration of the gene expression inductor, and agitation speed) are fundamental parameters that affect HA yield. The optimization of these parameters is fundamental for obtaining the highest HA amount from bacterial fermentation.

## 7. HA as a Powerful Bioactive Molecule

HA has been used for both medical and commercial applications thanks to its biocompatibility, biodegradability, and non-immunogenicity. Therefore, several HA-based products have been developed and are currently available on the market. HA showed its major potential for aesthetic applications but also was revealed to be promising in other biomedical fields. This is due to the fact that HA was revealed to be a versatile biomaterial with tunable properties that allowed the development of different kinds of devices (i.e., injectable formulations, gels, nanoparticles, sponges, and hydrogels) according to the injured tissue’s specific requirements. The main applications of HA as a powerful bioactive molecule are summarized in Figure 4.

### 7.1. Bioengineering Approaches

HA is often used as a biomaterial for several biomedical purposes. However, its rapid degradation and clearance from the body can limit its effectiveness in providing long-term therapeutic benefits [95,96]. Additionally, HA’s inherent low mechanical strength may restrict its use in load-bearing applications or environments where mechanical integrity is crucial [97,98,99].

Similarly, in cosmetic applications, HA is widely used for its ability to hydrate and plump the skin, reducing the appearance of wrinkles and fine lines [100,101]. However, the effects of HA treatments are typically temporary, as the injected or topically applied HA is gradually broken down and eliminated from the body [102,103,104]. This necessitates frequent reapplication or reinjection to maintain the desired cosmetic outcomes.

To address these limitations, researchers have been exploring various strategies to enhance the duration of HA’s effects, make the product more resistant to enzymatic degradation, and improve its mechanical integrity. These include chemical modifications of HA molecules and crosslinking between HA molecules or with other polymers [97] such as methyl cellulose [105,106,107,108,109,110], collagen [111,112,113,114,115,116,117], polyacrylic acid [118,119], and poly (lactic-co-glycolic acid) [120,121].

These approaches aim to extend the stability and persistence of HA within the body or improve its mechanical properties to better suit specific applications.

The presence of hydroxyl and carboxyl groups in the structure of HA enables easy chemical modifications, allowing the introduction of different functional groups such as thiols, haloacetates, aldehydes, dihydrazides, or carbodiimides. These modifications enable the crosslinking of HA, which is widely used in many injectable commercial products to enhance HA longevity and modify the viscoelastic properties of its aqueous solution. The crosslinking process is critical in determining the rheological characteristics of injectable products, allowing for their tailored application in specific areas and for specific purposes [122]. One of the most commonly used crosslinking agents in leading HA injectables on the market is 1,4-butanediol diglycidyl ether (BDDE). BDDE contains epoxy rings at both ends of its main chain that are susceptible to ring-opening reactions under alkaline conditions [123,124]. When BDDE is combined with HA in an alkaline environment, the epoxy rings react with the hydroxyl groups present on HA, forming strong ether bonds. This crosslinking mechanism enables covalent bonding between HA molecules, resulting in the crosslinking of hyaluronic acid [123,124].

This process provides stability, durability, and prolonged effects to the injected product. Overall, crosslinking of HA through the use of suitable agents such as BDDE plays a vital role in the development of injectable products with desired rheological properties, enabling their effective use in various medical and cosmetic applications [125].

On the other hand, one of the most popular chemical modifications of HA is the addition of acetyl groups on hydroxyls. Related to this, a study demonstrated that the acetylation of HA can enhance its chondrogenic potential in the context of bone tissue engineering [97]. Specifically, the presence of acetyl groups in acetylated HA scaffolds led to a decrease in negative charges within the molecule and helped balance the osmotic pressure, making it comparable to that of the natural ECM cartilage. This reasonable osmotic pressure contributed to the creation of an environment that was conducive to chondrogenesis [97,98].

Another study demonstrated the efficacy of HA acetylation on its bioavailability and stability compared to native HA [126]. In fact, the addition of acetyl groups to HA protected the molecule against enzymatic degradation [127,128].

Additionally, other chemical modifications can also affect HA applications. For example, thiolated crosslinked carboxymethylated HA has been proven effective in delivering drugs to the ocular surface when placed in the inferior fornix of the eye [129].

All of this suggests that HA can undergo chemical modifications or crosslinking processes based on its application and purpose.

### 7.2. HA-Based Injectables

HA-based injectables are known to be so effective for skin benefits [100,101] and viscosupplementation [130,131] that they have been on the market and clinically used for 15 years, as masterfully described by Vasvani et al. [132].

Several products have been developed and clinically tested in recent years. Due to the huge quantity of clinical studies performed on HA-based injectable products, only the most recent updates on clinically significant trials are cited in this review.

In an investigated report, 18 patients (25–40 years) were transdermally treated in front of the orbicularis muscle with 0.6–1 mL of Juvederm Ultra 3 (Allergan, Pringy, France), a product that contains 24 mg/mL of non-animal-derived HA 9% crosslinked with BDDE, lidocaine 0.3%, and phosphate buffer (pH 7.2).

The persistence of HA in the labial tissue and hyaluronidase activity were assessed in vitro and in vivo via magnetic resonance imaging (MRI) and histological and ultrastructural examination at 3 and 6months after treatment [17]. The MRI examination revealed the presence of HA filler in a clearly hyperintense area even after 6 months. The histological examination demonstrated tissue maturation after 3 months and the ability of HA to reorganize and integrate the extracellular matrix after 6 months. No correlation between hyaluronidase and HA was observed, so the obtained clinical results probably depended on systemic factors that may determine hyaluronidase activity and thus HA longevity [17].

Another recent controlled randomized trial evaluated the efficacy of the new monophasic HA filler Sardenyashape^®^ (RF Tech, Seoul, Republic of Korea) on nasolabial fold reduction [20] in 96 healthy patients (>30 years) with nasolabial folds. It was reported that Sardenyashape^®^, a crosslinked non-animal-based HA filler with BDDE (24 mg/mL), was more effective than biphasic HA in nasolabial fold reduction because of its ability to also fill the smallest spaces between collagen and elastin fibrils [20]. Unlike biphasic HA fillers, which consist of a heterogeneous combination of crosslinked and non-crosslinked HA particles, monophasic fillers are homogeneous and cohesive, making them more suitable for this specific purpose [20,133]. In addition, this product contains lidocaine 0.3% to reduce injection pain.

A further trial evaluated the efficacy and the safety of ART FILLER^®^ Volume (Laboratoires FILLMED, Pairs, France), an injectable crosslinked HA-based filler, in restoring facial volumes via supraperiosteal injection [9]. This product contains 25 mg/mL of non-animal-based HA, lidocaine 0.3%, and phosphate buffer (pH 7.2). Its distinctive feature is the Tri-Hyal Technology, which combines three different structures of HA (free HA, long, and very long chains). This combination of different HA structures contributes to the overall effectiveness and performance of the filler.

Hence, in this study, 98 patients were injected in different facial zones up to a maximum of two, and the efficacy and the safety of the product was evaluated and documented for 18 months [9].

The efficacy of HA-based dermal filler was also compared with other biopolymer-based products [15,16]. In particular, a clinical study compared the efficacy of four different fillers: three collagen-based fillers and an HA-based filler called Perlane^®^ (Medicis Pharmaceuticals, Scottsdale, AZ, USA) [16]. Although the four tested products led to a similar increase in lip volume, the presence of HA in the Perlane filler resulted in a greater increase than the others [10]. In particular, this product contains crosslinked and stabilized non-animal-based HA gel derived from bacterial fermentation and 0.3% lidocaine to reduce pain injection [16].

Another randomized controlled trial compared the efficacy of a collagen-based filler and an HA-based filler, the non-animal stabilized HA (NASHA) filler Restylane (Medicis Pharmaceutical Co., Scottsdale, AZ, USA), which contains 20 mg/mL NASHA dispersed in physiological saline solution [15].

In this case, the clinical outcomes of the two products in terms of effectiveness on skin benefits and safety were almost the same [15].

Regarding the effectiveness of HA injections in osteoarthritis, a recent study by Jin et al. (2020) [12] demonstrated the effectiveness of HA injections in treating the disease. The study utilized an injectable hydrogel composed of tyramine-conjugated HA and epigallocatechin-3-gallate to address inflammation and promote cartilage regeneration in osteoarthritic conditions. In vitro tests revealed that this product effectively shielded chondrocytes from the proinflammatory factor interleukin 1β (IL-1β) and facilitated chondrogenic regeneration. Furthermore, in vivo experiments demonstrated its efficacy in reducing cartilage loss [12].

Another effective study demonstrated the safety and effectiveness of Monovisc^®^ (Anika Therapeutics, Bedford, MA, USA) injections in providing symptomatic relief of pain caused by knee osteoarthritis (NCT00653432). This product contains partially crosslinked sodium hyaluronate (22 mg/mL) in a phosphate buffer solution.

In this trial, it was demonstrated that the treatment with HA caused a reduction in the pain score and other improvements in physical function (NCT00653432).

### 7.3. HA-Based Oral Formulations

It is known that the most-used HA products are HA fillers since these products represent the 80% of all fillers used for rejuvenation and volume correction [134]. Basically, more than 50% of the human body’s HA content is present in the skin [135,136], where it maintains extracellular spaces, preserves tissue hydration, facilitates the transport of ions and nutrients to cells [137,138], and is involved in cellular protection against radical damage [18].

A recent study analyzed the effects of Regulatpro Hyaluron (Dr. Niedermaier, Hohenbrunn, Germany), an oral-administered HA solution, on skin moisture content, elasticity, roughness, and wrinkle depths [18].

This product contains biotechnological-origin HMW HA (≥1 MDa) supplemented with biotin, vitamin C, copper, zinc, and natural silica [18]. In particular, the composition of this product is: water, REGULATESSENZ^®^ (fermented cascade concentrate of: water, lemons, figs, dates, common walnuts, soybeans, onions, coconuts, vegetable glycerin, celery, mung bean sprouts, acerola extract (17.5% vitamin C) 0.7%, artichokes, peas, millet, spice blend, turmeric, and saffron), plum concentrate, pear concentrate, horsetail extract 1%, L-ascorbic acid 0.375%, zinc gluconate 0.185%, hyaluronic acid 0.45%, copper sulfate 0.0049%, and D-biotin 0.00025%.

The administration of 20 mL/day to 20 women (aged 45–60 years) resulted in an increase in their skin elasticity and hydration and a decrease in skin roughness and wrinkle depths after 40 days [18]. Other promising results were obtained in a study conducted by Michelotti et al. in which the oral administration of 200 mg/day of ExceptionHYAL^®^ Star (Roelmi HPC, Origgio, Italy), a novel full-spectrum HA produced through advanced technologies and characterized by a large spectrum of molecular weights, for 28 days allowed the participants to have a better skin hydration level (+10.6%), a reduction in wrinkle depth (−18.8%) and volume (−17.6%), and an increase in elasticity and firmness (+5.1%) [19].

Thus, this product is a nutricosmetic created for skin health and aging that contains HA produced through advanced techniques using a large spectrum of MWs [19].

However, many studies demonstrated the efficacy of oral HA-based preparations for the treatment of knee osteoarthritis and urinary infections [13,29].

HA was combined with chitosan sulfate to orally treat knee osteoarthritis [13]. Thirty patients were randomly divided in two groups: a control group and the treatment group, in which the patients assumed Ialoral^®^ 1500 (PharmaSuisse Laboratories s.r.l., Milan, Italy) once a day for 4 weeks [13]. This product contains 120 mg of HA; 240 mg of chondroitin sulfate; and 300 mg of potassium, manganese and piperine.

It was observed that the treated patients had a significant reduction in clinical indices compared to the control subjects. In addition, the cytokine-dependent inflammatory activity also was decreased after the treatment, suggesting a potential anti-inflammatory effect of this preparation on cartilage and the synovial membrane [13].

The oral administration of HA has also been shown to be effective in preventing urinary tract infections in postmenopausal women [29]. In this case, two capsules containing HA, chondroitin sulfate, curcumin, and quercetin (both alone and in association with local estriol treatment) were administered daily for 15 days a month for 3 months, then one capsule was administered daily for 15 days a month for the next 9 months. This therapy was revealed to be effective in preventing recurrent urinary tract infections [29].

In addition, osteoarthritis can be treated by orally administering HA (NCT05683327). In particular, a recent uncompleted clinical trial aims to clarify the effect of UltraHA^®^ (Bloomage Biotechnology, Beijing, China), a product with a full spectrum of HA MWs, on knee joint conditions. Patients will be divided into three groups: the first one will receive 150 mg/day of HA, the second one will receive 80 mg/mL, while the third one will be the placebo group that will not receive HA. In this study, the investigators will evaluate the knee condition before and after the intervention. The groups to be compared are two different doses of UltraHA^®^ and a placebo (NCT05683327).

These results provided evidence supporting the effectiveness of HA as a dietary supplement. A recent study attempted to investigate the pharmacokinetic behavior of orally administered HA [139]. It was found that the HA underwent depolymerization in the gastrointestinal tract (GIT). This step is crucial for HA bioavailability, and this process is entirely dependent on the gut microbiota. In fact, in vivo experiments showed that the gut microbiota metabolizes more than 97% of the administered high-molecular-weight HA, with only approximately 0.2% being absorbed into the body [139]. In this way, the MW of HMW HA decreased, resulting in the formation of HA fragments of several kDa and unsaturated oligosaccharides; however, metabolomic analyses showed that only unsaturated oligosaccharides were available to the host [139]. In addition, in vitro studies confirmed the degradation of HA and identified several biologically active metabolites [139]. It was also demonstrated that the limited absorption and rapid metabolism of orally administered HA by the gut microbiota were not influenced by the molecular weight of the administered HA (ranging from 15 to 1600 kDa). This suggested that orally administered HA does not serve as nutrition for joints and skin, but rather its mechanism of action is likely due to the systemic regulatory function of HA or its metabolites [139].

### 7.4. HA-Based Drug Delivery Systems

In addition to oral formulations, HA’s potential also was investigated as a biomaterial for the manufacturing of bioactive drug delivery systems. In several works, empty and drug-loaded HA nanoparticles (HA-NPs) were obtained [14,28]. The interaction of the HA with its endogenous receptors (such as the CD44 receptor (CD44 R)) represented a very important advantage to be exploited [140].

In this regard, a study demonstrated the effectiveness of HA in atherosclerosis diagnosis and treatment [26]. Indeed, in this pathological condition the overexpression of stabilin-2 (another HA receptor) occurs in macrophages, smooth muscle cells, and endothelial cells of atherosclerotic plaques [141] and the high expression of CD44 R occurs on activated macrophages, which play fundamental roles in development of atherosclerotic plaques [142]. In vitro tests found that empty HA-NPs were selectively absorbed by cells expressing high levels of stabilin-2 and CD44 R through receptor-mediated endocytosis. In vivo tests also showed the uniform distribution of HA-NPs in atherosclerotic lesions through the active targeting mechanism because of the overexpression of the two HA receptors [26].

HA’s potential also was revealed for cancer treatment thanks to its interaction with the CD44 R. This receptor plays a crucial role in the invasion and metastatic processes of several human cancers [143]. Additionally, it is involved in mesenchymal epithelial transition [144,145], drug resistance [146], and resistance to chemo- and radiotherapy [147], and it is also a marker for cancer stem cells as well as gastric and breast cancers [148,149,150]. For this, Zhang et al. developed a cleavable PEGylated HA nano-drug delivery system in order to improve the active tumor targeting, tumor cell uptake efficiency, and circulation time of doxorubicin in vitro and in vivo [16]. In addition, in this case the interaction between HA and CD44 R was exploited, demonstrating the ability of HA-based nanoparticles to selectively target murine colon carcinoma cells that were CD44 positive. In this way, the circulation time of the antitumor drug loaded in HA-NPs and the tumoral tissue targeting were significantly increased. Consequently, the toxicity induced by the doxorubicin was decreased [23].

The interaction between HA and CD44 R also was exploited in the formulation of Oncofid-PB (Fidia Farmaceutici S.p.A., Abano Terme, Italy), a drug composed of paclitaxel bioconjugated with HA, which has been found to be effective in the treatment of intravesical cancer [24]. This product contains biotechnological-sourced HA tetrabutylammonium salt at 200 KDa [24].

In vitro studies demonstrated its superior cytotoxic activity (up to 800-fold) compared to unconjugated paclitaxel [24]. A phase I exploratory, open-label, single-arm, multicenter study showed excellent safety and tolerability of Oncofid-PB in subjects with intravesical cancer who were unresponsive or intolerant to Bacillus Calmette-Guérin (BCG) therapy [24]. Additionally, Oncofid-PB is currently undergoing a phase III single-arm, multicenter, international study in order to evaluate its efficacy and safety in patients unwilling or unfit to undergo radical cystectomy (NCT05024773).

Empty HA-NPs were also revealed to be effective for the treatment of osteoarthritis [14]: their cellular uptake was increased by the ectopic expression of CD44 R, resulting in their internalization. Moreover, the NPs were advantageously modified in order to resist against hyaluronidase degradation and to present a long-term retention ability in knee joint [14].

Empty HA-NPs can also be a viable alternative for psoriasis treatment [28]. It was recently demonstrated in murine models that HA-NPs were able to penetrate inflamed skin selectively target dermal macrophages, suppress the proinflammatory response, and restore the skin barrier at the same time [28]. Even more recently, a transdermal ginsenoside Rg3 (a molecular compound with anti-inflammatory and immunoregulatory functions) delivery system based on liposomes in HA-based microneedles was developed to provide efficient site-specific drug delivery, controlled drug release, and enhanced permeability. Reductions in the thickness of thorny skin and the production of inflammatory cytokines along with an increase in drug retention without causing skin irritation were observed [27].

Liposomes functionalized with HA have also been used in a tumor-targeting drug delivery system. In a study by Deng et al., HA was used to functionalize liposomes for doxorubicin administration in melanoma cells [22]. In this way, the antitumoral drug was more easily internalized by cancer cells. It was accumulated near the nucleus; consequently, the drug concentration at the tumor site was found to be higher. In addition, a better drug circulation time and a reduction in side effects were registered [22].

The results described so far on the use of HA in drug delivery are promising and show that research is well on its way in this direction. However, at the same time it is evident that these systems are widely described only in vitro and in vivo in animals. For human testing, in fact, the steps to be taken are still numerous; in these cases, it is necessary to evaluate not only the effectiveness and safety of the HA but also of the crosslinking agents, additives, and other NPs components.

### 7.5. Other HA-Based Products

It was previously described that HA plays a crucial role in maintaining skin hydration and elasticity, and its decrease contributes to the visible effects of aging on the skin. Therefore, the primary strategy to mitigate skin aging involves the application of HA both externally and internally to ensure its effective penetration into the epidermis and dermis.

Some researchers developed an HA matrix extracted and purified from rooster combs composed of 67% HA with an average MW of 1.3 MDa, 12% sulfated GAGs, 17% proteins, and water [37]. This product showed its effectiveness in fibroblast and keratinocyte regeneration as well as its moisturizing, antiaging, and antioxidant effects [37].

Another recent study evaluated the efficacy of silk fibroin/HA/sodium alginate scaffolds in skin repair [38]. In vitro experiments demonstrated that these scaffolds exhibited improved mechanical properties as well as an appropriate swelling capacity and facilitated the proliferation and adhesion of NIH-3T3 cells. In vivo studies revealed that these scaffolds accelerated the wound-healing process and provided an advantageous environment for dermal repair with a suitable degradation rate, leading to significant skin reconstruction [38].

HA also can be used to develop hydrogel patches useful in tissue engineering [32]. Choi et al. (2020) [32] developed osteoconductive pyrogallol-conjugated HA hydrogel patches to promote effective bone formation. In particular, hydroxyapatite, whitlockite, and bone morphogenetic protein-2 were incorporated in these patches, which exhibited improved degradative behaviors and mechanical properties, enhanced osteogenic differentiation of human stem cells in vitro, and effective formation of mineralized bone in vivo [32].

## 8. Conclusions

As a biocompatible and biodegradable compound with exceptional properties, HA holds great potential as a bioactive molecule for various physiological and pathological conditions. As a result, there is a growing interest in commercial endeavors to develop new and more efficient production processes and therapeutic products. Extensive industrial and scientific research is being conducted to optimize existing methods and achieve high-quality HA products at affordable costs.

This review article primarily focused on the HA biosynthetic pathway and highlighted that the yield of HA is limited by several factors because many metabolic intermediates are required for other cellular processes. However, by optimizing the culture conditions, both the yield and molecular weight (MW) of the final product can be adjusted.

While natural HA-producing microorganisms have shown potential, they also present certain challenges, particularly those related to the presence of endotoxins in the final product. This issue has been addressed by employing recombinant systems. Various bacteria such as *E. coli*, *B. megaterium*, and *B. subtilis* have been genetically engineered due to their safety, ability to produce significant amounts of HA, and ease of manipulation in the laboratory.

The article discussed the outcomes of HA-based products in the health-related sector over the past decade, highlighting how HA has emerged as a bioactive molecule with substantial potential in various biomedical aspects. These range from skin benefits to cancer therapy and from inflammatory conditions to chronic pathologies. Ongoing research continues to explore these areas, but certain limitations need to be overcome, starting with the high production costs of HA. Additionally, the potential occurrence of allergic reactions due to microbial proteins in the final product and the presence of endotoxins pose further challenges. Limited clinical trials, which require considerable time, add to the complexity.

In conclusion, the development of innovative HA production systems is a crucial objective to meet the increasing demand for HA at sustainable costs. Simultaneously, understanding new bioactive HA pathways coupled with conducting clinical trials of HA-based products drives biomedical and pharmaceutical research in this field.

## Figures and Tables

**Figure 1 ijms-24-10296-f001:**
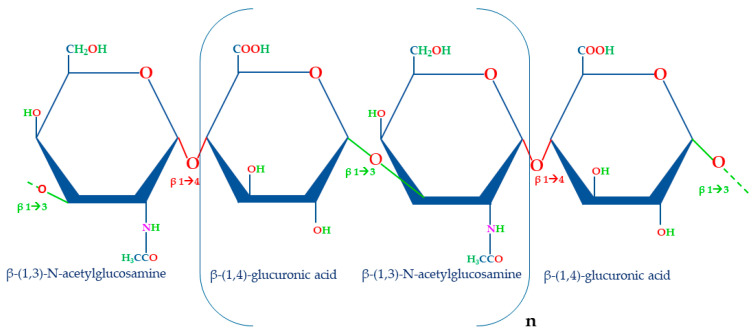
Structure of disaccharide repeating unit of HA. The unit of HA is composed of β-(1,4)-glucuronic acid and β-(1,3)-N-acetylglucosamine linked together by β-1,3 and β-1,4 glycosidic bonds. The molecular weight of this molecule depends on the number of repetitions of the disaccharide unit (n).

**Figure 2 ijms-24-10296-f002:**
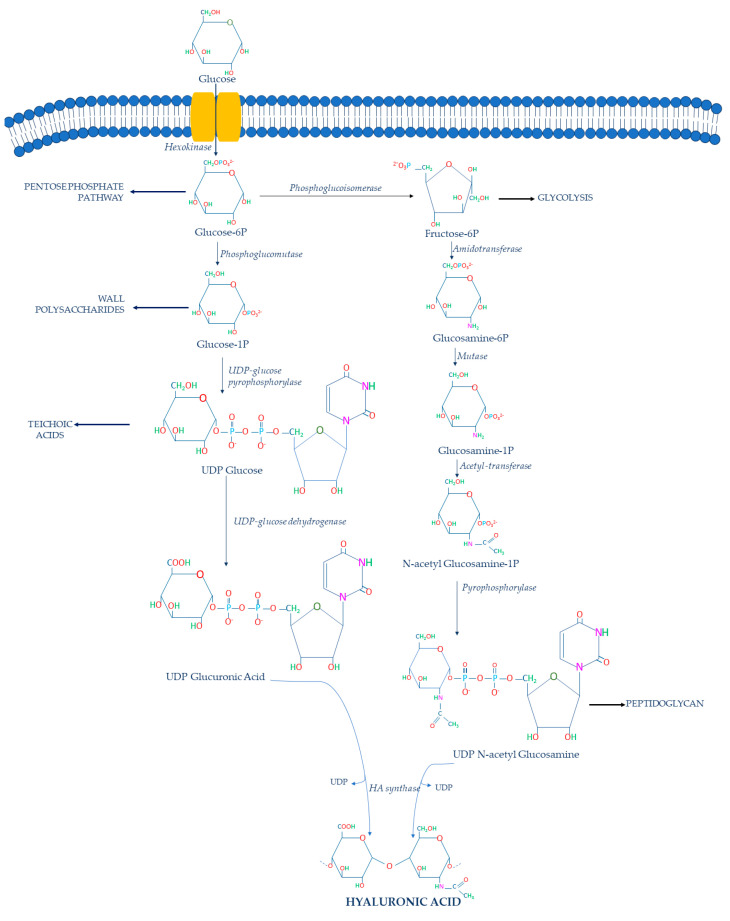
HA biosynthetic pathway in *S. zooepidemicus*.

**Figure 3 ijms-24-10296-f003:**
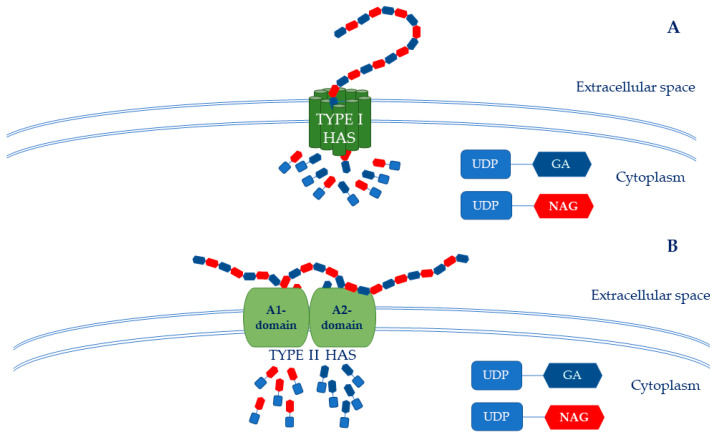
HAS structure. (**A**) The Type I HAS is an integral membrane protein that catalyzes the adding of UDP-GA and UDP-NAG to the HA growing chain. (**B**) The Type II HAS is a membrane-associated protein composed of A1 and A2 domains that work together as a single peptide. The A1 domain is a β-1,4-N-acetylglucosaminil-transferase, while the A2 domain is a β-1,3-glucuronyl-transferase.

**Figure 4 ijms-24-10296-f004:**
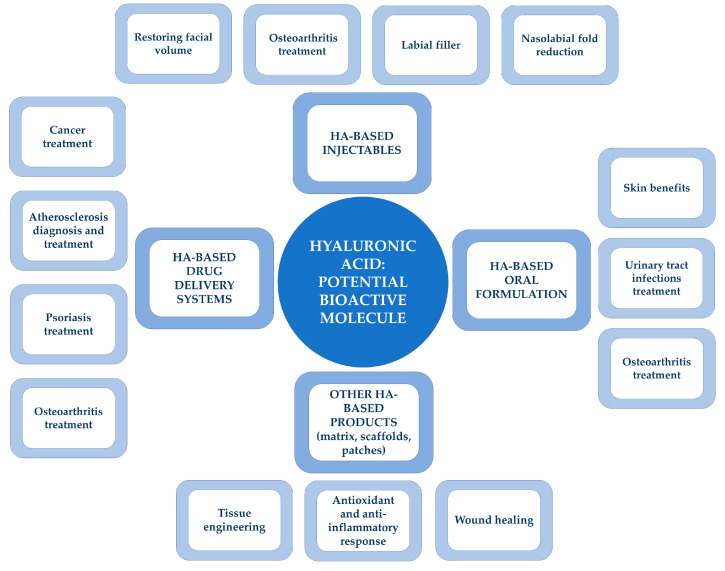
HA: a powerful bioactive molecule. Schematical representation of the most important HA applications in the health-related sector.

**Table 1 ijms-24-10296-t001:** Advantages and limitations of HA extracted from different sources.

HA Source	Advantages	Limitations	Refs.
Animal sources (rooster combs, umbilical cords, bovine vitreous humor)	High MW HA, natural-like product	High extraction costs, presence of animal proteins, high polydispersity, contamination risks	[83,84,85]
Natural producer microorganisms (groups A and C streptococci and *Pasteurella multocida*)	High MW HA, high yields, easy production method, low polydispersity	Presence of endotoxins, high purification costs	[62,74,86]
Engineered safe microorganisms (i.e., *E. coli*, *B. subtilis*, *B. megaterium*)	High MW, high yields, safety of the final product, low polydispersity	Risk of contamination with bacterial proteins and nucleic acid	[81]

## Data Availability

Not applicable.

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
