# Peer review of "Hyaluronic Acid: A Powerful Biomolecule with Wide-Ranging Applications—A Comprehensive Review"

_ijms, 2023, doi:10.3390/ijms241210296_

Round 1
Reviewer 1 Report
The manuscript titled "Hyaluronic acid: a powerful biomolecule with wide-ranging applications - a comprehensive review" provides a comprehensive understanding of the applications of hyaluronic acid. The article is well-written, with good English language proficiency. However, I have identified a few areas that require attention to improve the manuscript and align it with the scope of the special issue.
Major Comments:
Lack of comprehensive coverage:
While the manuscript provides insightful information on three key biomedical applications of hyaluronic acid, namely injectable fillers, oral supplementation, and drug delivery systems, it lacks detailed descriptions of two important aspects: [1] the antioxidant and anti-inflammatory/immunomodulatory efficacy of hyaluronic acid and [2its applications in wound healing and tissue engineering. To ensure the manuscript meets the scope of the special issue, I strongly recommend incorporating these additional areas of research.
Incorporation of recent bioengineering approaches:
An important aspect missing from the manuscript is the recent bioengineering approaches that involve chemical modifications of hyaluronic acid to enhance its bioavailability. For example, acetylated HA and methylated HA are two such modifications that have shown promise in improving the properties of hyaluronic acid. It is suggested to include a section discussing these recent bioengineering approaches and their potential impact on hyaluronic acid's applications. Provide an overview of the modifications, their effects on bioavailability, and any relevant research findings or studies supporting their efficacy.
Mechanism of action of orally administered hyaluronic acid (Lines 430-469):
In the section discussing orally administered hyaluronic acid (Lines 430-469), it would greatly benefit the readers if the manuscript includes evidence or, at the very least, speculations regarding the mechanisms of action underlying its effects. Presenting potential mechanisms or hypotheses can strengthen the scientific basis of the manuscript and provide valuable insights to the readers. It is recommended to support the statements with relevant research findings or cite references to support the proposed mechanisms.
Minor Comments:
Clarification on "biphasic HA" (Line 402):
Line 402 mentions the term "biphasic HA" without a clear explanation. It is crucial to provide a concise and precise definition or description of what "biphasic HA" refers to in order to enhance the reader's understanding. Consider providing a brief explanation or citing relevant literature to clarify this term.
While the manuscript demonstrates a good command of the English language, it would be helpful to review the text for any minor grammatical or typographical errors.
Overall, I find the manuscript to be well-written and comprehensive. By addressing the aforementioned points, the manuscript can be further enhanced to meet the requirements of the special issue. I recommend the authors consider the suggestions and revise the manuscript accordingly carefully.
Author Response
Reviewer #1:
The manuscript titled "Hyaluronic acid: a powerful biomolecule with wide-ranging applications - a comprehensive review" provides a comprehensive understanding of the applications of hyaluronic acid. The article is well-written, with good English language proficiency. However, I have identified a few areas that require attention to improve the manuscript and align it with the scope of the special issue.
Major Comments:
Lack of comprehensive coverage:
While the manuscript provides insightful information on three key biomedical applications of hyaluronic acid, namely injectable fillers, oral supplementation, and drug delivery systems, it lacks detailed descriptions of two important aspects: [1] the antioxidant and anti-inflammatory/immunomodulatory efficacy of hyaluronic acid and [2] its applications in wound healing and tissue engineering. To ensure the manuscript meets the scope of the special issue, I strongly recommend incorporating these additional areas of research.
response: we have introduced the paragraph 7.5 (lines 712-735). See also lines 501-512.
Incorporation of recent bioengineering approaches:
An important aspect missing from the manuscript is the recent bioengineering approaches that involve chemical modifications of hyaluronic acid to enhance its bioavailability. For example, acetylated HA and methylated HA are two such modifications that have shown promise in improving the properties of hyaluronic acid. It is suggested to include a section discussing these recent bioengineering approaches and their potential impact on hyaluronic acid's applications. Provide an overview of the modifications, their effects on bioavailability, and any relevant research findings or studies supporting their efficacy.
response: we have introduced the paragraph 7.1 (lines 409-469).
Mechanism of action of orally administered hyaluronic acid (Lines 430-469):
In the section discussing orally administered hyaluronic acid (Lines 430-469), it would greatly benefit the readers if the manuscript includes evidence or, at the very least, speculations regarding the mechanisms of action underlying its effects. Presenting potential mechanisms or hypotheses can strengthen the scientific basis of the manuscript and provide valuable insights to the readers. It is recommended to support the statements with relevant research findings or cite references to support the proposed mechanisms.
response: we have introduced the mechanism of action (lines 576-591).
Minor Comments:
Clarification on "biphasic HA" (Line 402):
Line 402 mentions the term "biphasic HA" without a clear explanation. It is crucial to provide a concise and precise definition or description of what "biphasic HA" refers to in order to enhance the reader's understanding. Consider providing a brief explanation or citing relevant literature to clarify this term.
response: we have clarified the term “Biphasic HA”, lines 476-480.
Reviewer 2 Report
The review by Iaconisi G.N. et al. focuses on hyaluronic acid, processes for its production, and medical/cosmetological applications. The authors first briefly describe the structure and basic properties of hyaluronic acid, then the possible routes of synthesis and applications as injectables, drug delivery systems, and oral formulations. The disadvantages of the review are its descriptive form without data analysis and generalizations. The review does not describe the salt form of hyaluronic acid, cross-linking of hyaluronic acid, and an analysis of the effect of HA-based products' composition on their properties. There are no generalizations in the form of concentrations, molecular weights, crosslinkers, and degree of crosslinking to be used. The review needs to be revised. In addition, I do not feel that this article is appropriate for the special issue “Cell Metabolism and Small Natural Compounds”, as hyaluronic acid is a giant natural compound.
Specific comments are as follows.
Title: “A COMPREHENSIVE REVIEW”. This review contains only 88 references on such an active topic as hyaluronic acid, which is not enough to be "comprehensive". Maybe the authors can clarify the topic/type of the review.
Line 60: “for pharmacological and biomedical applications [9-26].” It is inappropriate to give 17 references at once without explaining them. The authors are encouraged to list specific pharmacological and biomedical applications with specific references for each area.
Line 82: “The key words used were ‘hyaluronic acid’”. The authors should search for the words "hyaluronan" and "hyaluronate" for a complete review. In addition, the review does not mention anything about sodium hyaluronate, which is often used instead of hyaluronic acid. The authors should write something about this. For example, when native hyaluronic acid is used and when its salt does.
Line 386: “Several products have been developed and clinically tested in recent years. Due to the huge quantity of clinical studies performed on HA-based injectable products, only the most recent updates on clinically significant trials have been cited in this review.” A review of clinical studies is a good idea. However, a specific feature of HA-based injectables is that the hyaluronic acid is cross-linked, usually using 1,4-butanediol diglycidyl ether. Cross-linking is performed to change the viscoelastic properties of aqueous solutions of hyaluronic acid, as the rheology of injectable products determines the possibility of their use for specific zones and applications (see, e.g., 10.1007/s00397-016-0913-z). This point should be noted.
Lines 390, 400, 404, 414, 419, 423, 437, 443, 451, 464, 499: “Juvederm (Ultra 3, Allergan, Pringy, France)”, “new monophasic HA filler, Sardenyashape® (RF Tech, Seoul, Republic of Korea)”, “ART FILLER® Volume (Laboratoires FILLMED, Pairs, France), an injectable HA-based filler”, “an HA-based filler, Perlane ® (Medicis Pharmaceuticals, Scottsdale, Arizona)”, “a HA based filler, NASHA filler Restylane (Medicis Pharmaceutical Co., Scottsdale, Arizona)”, “Monovisc® (Anika Therapeutics, Bedford, Massachusetts) injections”, “Regulatpro Hyaluron (Dr. Niedermaier,Hohenbrunn, Germany), an oral administered HA solution”, “ExceptionHYAL® Star (Roelmi HPC, Origgio, Italy), a supplement containing hyaluronans of different MW”, “Ialoral® 1500(PharmaSuisse Laboratories Srl, Milan, Italy)”, “UltraHA® (Bloomage biotechnology, China)”, “Oncofid-PB (Fidia Farmaceutici S.p.A., Abano Terme, Italy), a drug composed of paclitaxel conjugated with HA”. The authors provide the commercial brands but do not indicate the composition of these systems. After all, this is a scientific review rather than advertising of products, and therefore the composition of these products (concentration of HA, its molecular weight, cross-linking, and so on), their comparison, and effectiveness are of scientific value and should be presented.
The English language is fine and requires almost no corrections.
Author Response
Reviewer #2:
The review by Iaconisi G.N. et al. focuses on hyaluronic acid, processes for its production, and medical/cosmetological applications. The authors first briefly describe the structure and basic properties of hyaluronic acid, then the possible routes of synthesis and applications as injectables, drug delivery systems, and oral formulations. The disadvantages of the review are its descriptive form without data analysis and generalizations. The review does not describe the salt form of hyaluronic acid, cross-linking of hyaluronic acid, and an analysis of the effect of HA-based products' composition on their properties. There are no generalizations in the form of concentrations, molecular weights, crosslinkers, and degree of crosslinking to be used. The review needs to be revised. In addition, I do not feel that this article is appropriate for the special issue “Cell Metabolism and Small Natural Compounds”, as hyaluronic acid is a giant natural compound.
Specific comments are as follows.
Title: “A COMPREHENSIVE REVIEW”. This review contains only 88 references on such an active topic as hyaluronic acid, which is not enough to be "comprehensive". Maybe the authors can clarify the topic/type of the review.
response: According to the Reviewer suggestion, we have revised the manuscript, introduced new paragraphs to make the review more understandable and appropriate references.
Line 60: “for pharmacological and biomedical applications [9-26].” It is inappropriate to give 17 references at once without explaining them. The authors are encouraged to list specific pharmacological and biomedical applications with specific references for each area.
we revised the manuscript following the given suggestions, introduced new paragraphs to make the review more understandable. This also resulted in an increase in the number of references.
response: we have explained the different applications and for each one we have listed specific references (lines 59-66).
Line 82: “The key words used were ‘hyaluronic acid’”. The authors should search for the words "hyaluronan" and "hyaluronate" for a complete review. In addition, the review does not mention anything about sodium hyaluronate, which is often used instead of hyaluronic acid. The authors should write something about this. For example, when native hyaluronic acid is used and when its salt does.
response: we also referred to the term sodium hyaluronate in the revision of the manuscript, indicating when the salt is used as a substitute for acid.
Line 386: “Several products have been developed and clinically tested in recent years. Due to the huge quantity of clinical studies performed on HA-based injectable products, only the most recent updates on clinically significant trials have been cited in this review.” A review of clinical studies is a good idea. However, a specific feature of HA-based injectables is that the hyaluronic acid is cross-linked, usually using 1,4-butanediol diglycidyl ether. Cross-linking is performed to change the viscoelastic properties of aqueous solutions of hyaluronic acid, as the rheology of injectable products determines the possibility of their use for specific zones and applications (see, e.g., 10.1007/s00397-016-0913-z). This point should be noted.
response: the cross-linking has been introduced in 7.1 paragraph (see lane 397).
Lines 390, 400, 404, 414, 419, 423, 437, 443, 451, 464, 499: “Juvederm (Ultra 3, Allergan, Pringy, France)”, “new monophasic HA filler, Sardenyashape® (RF Tech, Seoul, Republic of Korea)”, “ART FILLER® Volume (Laboratoires FILLMED, Pairs, France), an injectable HA-based filler”, “an HA-based filler, Perlane ® (Medicis Pharmaceuticals, Scottsdale, Arizona)”, “a HA based filler, NASHA filler Restylane (Medicis Pharmaceutical Co., Scottsdale, Arizona)”, “Monovisc® (Anika Therapeutics, Bedford, Massachusetts) injections”, “Regulatpro Hyaluron (Dr. Niedermaier,Hohenbrunn, Germany), an oral administered HA solution”, “ExceptionHYAL® Star (Roelmi HPC, Origgio, Italy), a supplement containing hyaluronans of different MW”, “Ialoral® 1500(PharmaSuisse Laboratories Srl, Milan, Italy)”, “UltraHA® (Bloomage biotechnology, China)”, “Oncofid-PB (Fidia Farmaceutici S.p.A., Abano Terme, Italy), a drug composed of paclitaxel conjugated with HA”. The authors provide the commercial brands but do not indicate the composition of these systems. After all, this is a scientific review rather than advertising of products, and therefore the composition of these products (concentration of HA, its molecular weight, cross-linking, and so on), their comparison, and effectiveness are of scientific value and should be presented.
response: we have indicated the composition of these systems.
Round 2
Reviewer 2 Report
The authors have greatly improved their review. Now it is indeed a comprehensive review that merits publication.
The English language is fine and requires almost no corrections.